# Gastrointestinal Stromal Tumour with Liver Metastasis Presenting as Gastric Cancer

**DOI:** 10.3390/diagnostics13030376

**Published:** 2023-01-19

**Authors:** Yongwei Xu, Bacui Zhang, Jing Wang

**Affiliations:** Department of Gastroenterology, Songjiang Hospital, School of Medicine, Shanghai Jiaotong University, No. 746, Zhongsanzhong Road, Songjiang District, Shanghai 201600, China

**Keywords:** gastrointestinal stromal tumours, tumour metastasis, gastric cancer

## Abstract

Gastrointestinal stromal tumour (GIST) is a malignant tumour of the gastrointestinal lobe tissue, which mostly occurs in the gastrointestinal tract. Clinical manifestations can range from being benign to malignant. It mainly occurs in the gastric and small intestine. It may also develop in the colon, oesophagus, and bowel membranes, or outside the gastrointestinal tract and intestines. The pathological diagnosis of GIST depends on morphological measurements and immunohistochemistry. We report an interesting case in which the patient’s gastroscopy indicated gastric malignant tumours, and the results of the contrast-enhanced computed tomography (CT) of the upper abdomen showed malignant stomach tumour accompanied by liver metastasis. After the patient knew about this diagnosis, she wanted to give up treatment. Finally, the gastric biopsy suggested positive CD34, CD117, DOG1, and Ki-67, which supported the diagnosis of GIST. We hope that, through this case, we could improve clinicians’ understanding of GIST and improve its diagnosis and treatment.

Gastrointestinal stromal tumours (GISTs) have an incidence of ~1.2 per 10^5^ individuals per year in most countries [1]. GISTs are sarcomas that mostly derive from precursors of the interstitial cells of Cajal (ICC), the pacemaker cells of the gastrointestinal tract responsible for its motility. GIST is the most frequent of all sarcomas [2,3]. GIST is a heterogeneous group of tumours that includes a variety of molecular entities with usually mutually exclusive activating oncogene mutations, mostly KIT or PDGFRA mutations [4]. GISTs can occur anywhere along the GI tract and are most commonly found in the stomach (55%), followed by the small bowel (32%), large bowel (6%) and oesophagus (1%). Six percent of GIST cases occur outside of these locations [5]. The clinical presentation of GIST is that of a gastric or bowel tumour. Bleeding, pain and/or obstruction are frequent symptoms at presentation. 

It is hoped that, through this case, clinical workers can increase their understanding of GIST and the performance of endoscopic imaging.

GIST is the most common tumour of mesenchymal tissue origin originating from the digestive tract, accounting for 0.1–3% of gastrointestinal malignancies. GIST usually has no specific symptoms and is often found incidentally during physical examination or surgical treatment of other diseases. Some patients take gastrointestinal bleeding as the primary symptom, mainly manifested as abdominal pain, abdominal mass, nausea, vomiting, fatigue, and weight loss. This patient has had intermittent abdominal pain and felt discomfort for two years, and the patient had not paid much attention to it. The gastroscopy of the patient suggested the possibility of gastric cancer. In the imaging diagnosis, gastric malignancy with liver metastasis was also presented. The patient almost gave up treatment. Finally, with the pathological diagnosis of GIST, the patient retains the hope of treatment.

A 76-year-old female patient had intermittent abdominal pain and discomfort two years ago. It was located under the sword process, showing a hidden pain, and it was sustainable. Pain after eating a meal was significantly reduced. The patient came to our hospital for a gastroscopy. The results of the gastroscopy are shown in Figure 1, which suggest gastric stomach cancer. After the patient was admitted to the hospital, the upper abdomen was imaged using a contrast-enhanced computed tomography (CT) (Figure 2).

GIST is the most common source of tissue from the digestive tract, accounting for 0.1% to 3% of gastric and intestinal malignant tumours [7]. GIST usually has no specific symptoms. It is often discovered during medical examination or surgery during the treatment of other diseases. Some patients take gastrointestinal bleeding as the primary symptom, mainly manifested as abdominal pain, abdominal blocks, nausea, vomiting, and fatigue. For this case example, it is repeated abdominal pain and disease-free gastrointestinal symptoms.

The diagnosis and follow-up of GISTs mainly involve MRI and CT. These methods can accurately determine the size, location, infiltration depth, and non-distant metastasis of tumours [8]. However, the ultimate diagnosis still needs a pathological diagnosis.

CD117 is the most characteristic immunomarker of GIST. Its combination with DOG1 and CD34 testing has been used as the main indicator of GIST diagnosis, but in recent years, about 10% of cases of CD117 are negative or weak. Therefore, combining with C-Kit/PDGFRO gene mutation test can avoid the omission of patients with CD117 negative GIST. For patients who cannot have surgery, it is of great significance to perform a fine-needle puncture biopsy [9].

GISTs are commonly encountered tumours in routine practice. In the main tumour, the morphology of spindle, epithelioid, or mixed cells are well recognised, along with mutations of c-kit. In this article, the results of the H&E staining for histology by GIST microscopy reveal that it is mainly composed of spindle and epithelioid-like cells (Figure 3). CD117 is the most characteristic diagnostic marker of GIST. Its combination with DOG1 and CD34 testing has always been used as the main indicator of GIST diagnosis [10,11]. Antonescu CR et al. showed that GISTs usually exhibit monotonous morphology characterized by uniform cells with minimal cytologic atypia, rare mitoses, and immunoreactivity for CD117, which is encoded by the KIT gene [11]. Approximately 95% of classic GISTs typically express CD117 and CD34 in about 60–70% of the lesions [12]. In addition, Novelli M et.al showed that DOG1 and CD117 are the most sensitive and specific antibodies used in GIST diagnosis [13]. The expression of Ki-67 has significant value in predicting the risk grade and prognosis of GISTs. GISTs present a risk of recurrence and metastasis, so patients may need a close follow-up [14].

Most of the CD117 and CD34 of the tumour immunohistochemical are positive. In CD117 negative cases, combining with DOG1 testing can improve the accuracy of diagnosis of GIST. For some CD117 and DOG1 negative cases, PDGFR -α and C -KIT gene mutation detection must be performed. For this patient, the immunohistochemical results for CD34, CD117, and DOG1 are positive (Figure 4).

In the endoscopic inspection, we found that the surrounding area appears to have a submucosal tumour (SMT)-like shape. Gastric SMT refers to the submucosal bulge of the stomach. It is the bulge lesion of the stomach seen under the gastroscope. Because the bulge originates from the submucosa, the colour and structure of the surface are consistent with the surrounding gastric mucosa. Under the endoscopic examination, we see the portion of the stomach occupied by tumours and highlight the stomach cavity, which shows unevenness; thus, a diagnosis of gastric malignant tumours is made (Figure 1). The finding of a submucosal bulge in the stomach requires ultrasound gastroscopy to identify the exact source of the submucosa and to determine its nature. Finally, we performed a pathological examination to diagnose GIST (Figure 4).

At present, surgical resection is the gold standard for the treatment of limited primitive GIST [16]. Whether to completely remove the tumours is the main factor affecting the prognosis. According to YAN JZ et al.’s research, a tumour needs to be removed as soon as possible for those who have surgery, and its 5-year survival rate can reach 42% [17]. For patients with irregularity, metastasis, or recurrence, tyrosine kinase inhibitors can be used to improve the prognosis of patients with GIST. The use of imatinib in the treatment of GIST has achieved good results. 

Once GIST is diagnosed, it should be treated early. Surgical resection combined with molecularly targeted drugs, such as imatinib, is the current consensus in GIST therapy. A surgical procedure is the only cure. As a first-line adjuvant drug for GIST patients, imatinib has remarkable efficacy and can significantly delay postoperative recurrence and prolong survival.

Because of its relatively clear pathogenic mechanism, imatinib can target the pathological activation of tyrosine kinase caused by Kit/PDGFRA mutation. As a first-line drug treatment for non-surgical resection of patients with GIST, imatinib opens a new era of targeted therapy for stromal tumours. However, tumours are not completely eradicated and often recur because of the interruption of medication. Approximately 14% of GIST cases develop secondary resistance, and, in this study, the immune checkpoint inhibitors developed mainly include programmed cell death-1/programmed cell death ligand-1, (PD1/PD-L1) and cytotoxic T-lymphocyte-associated protein-4 (CTLA4) [18]. Immunotherapy, as an emerging direction for tumour treatment, has also gradually developed in the treatment regimen of GIST patients.

## Figures and Tables

**Figure 1 diagnostics-13-00376-f001:**
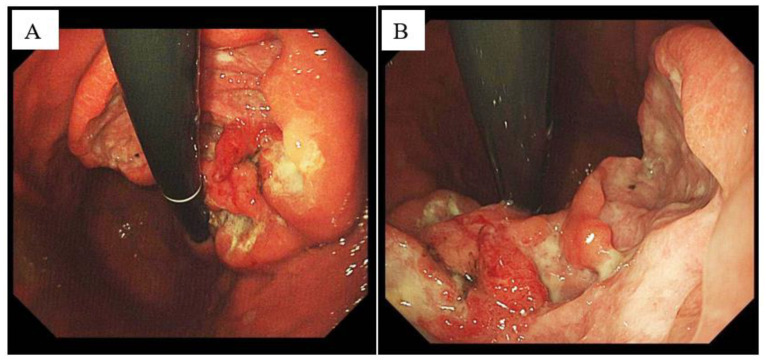
The patient’s gastroscopic results before admission. The stomach can be seen in the populous lesions of the volcanic mouths (**A**). It can be seen that the bottom is uneven and there is pus (**B**). Endoscopy is important for the diagnosis of GIST. It shows that non-specific smooth processes cover the normal mucosa, with ulceration, necrosis, and bleeding features. It can be seen from the gastroscopy that the mucosal folds are at the backbone, which are orange red, there is a small bend in the middle of the middle, and the back walls have huge placement lesions. The bottom of the bottom is a selfie, the surrounding embankment bulges, and the biopsy is tough.

**Figure 2 diagnostics-13-00376-f002:**
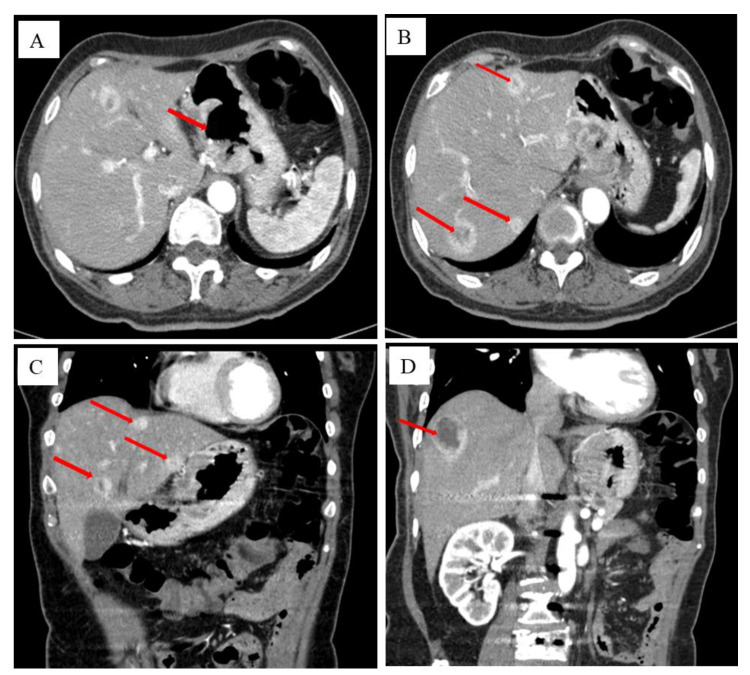
The patient’s results from the CT examination after admission. After the patient was admitted to the hospital, the transverse section of the upper abdomen was imaged using a contrast-enhanced CT, showing gastrointestinal lesions, which are like a volcanic mouth (red arrow) (**A**). The transverse section indicates gastric cancer metastases to the liver (red arrow) (**B**). The median sagittal section shows that gastric cancer is accompanied by liver metastasis (red arrow) (**C**). The coronal section shows that gastric cancer is accompanied by liver metastasis (red arrow) (**D**). The patient is diagnosed with malignant tumours of the gastric gastrointestinal side and ulcers, lymphatic metastasis to the liver and gastrointestinal ligament area, and multiple metastases in the liver. QIAN XH et al. find that GIST is generally considered to originate from the Khalk cell ICC. Most of the GIST occurs in the stomach, followed by small intestines, the colon, and the oesophagus [6]. Malignant GISTs can be transferred to the liver through blood circulation, and those that occur in the liver are basically the gold standard for transferring liver GISTs.

**Figure 3 diagnostics-13-00376-f003:**
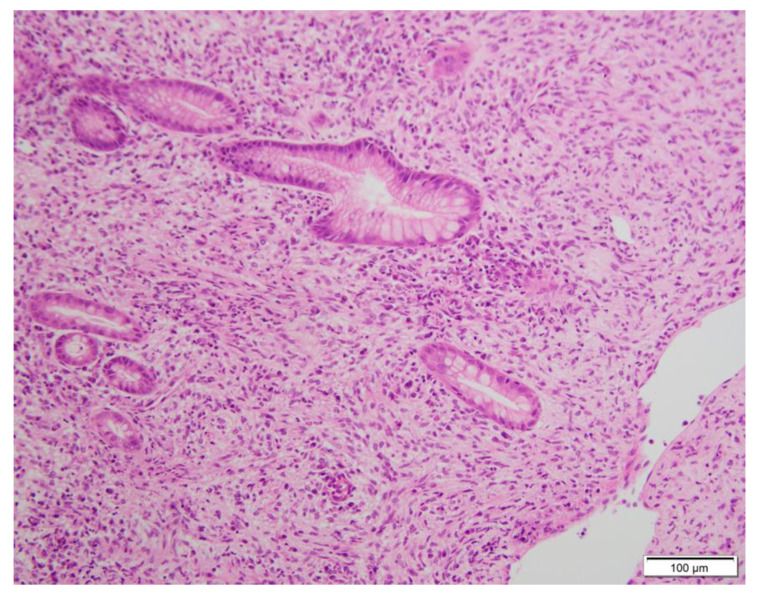
Gastroscopy biopsy tissue H&E stain can be seen from scattering cells. (H&E stain, ×200). According to the form of tumour cells, GIST is divided into three major subtypes: spindle cell, epithelial sample, and mixed type. The rare types include de-differentiated types. Spindle cell accounts for 50% to 70%. It is mainly composed of scattering cells with relatively consistent morphology. The density, heterogeneity, and nuclear split elements of tumour cells vary for each case. Some GIST cases can be seen on the nuclear end. Slipper cells are mostly arranged or intertwined, and sometimes there are a variety of arrangements, such as organs, fake chrysanthemum-shaped groups, or fences [15].

**Figure 4 diagnostics-13-00376-f004:**
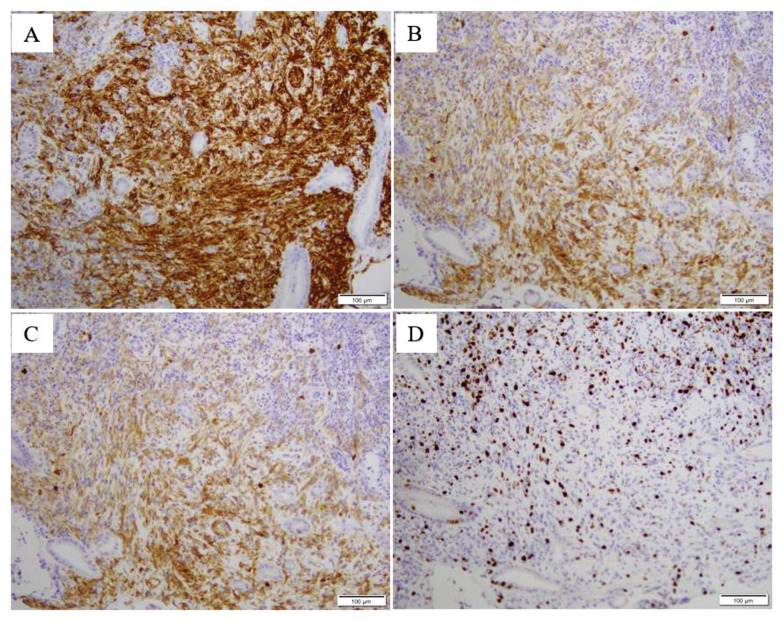
CD34 (**A**), CD117 (**B**), DOG1 (**C**), and Ki-67 (**D**) of the gastroscopic biopsy after admission. The preoperative diagnosis of GIST clinically depends on the abdomen CT, such as CT has found that tumours grow outside or between the cavity, with large diameter, unclear boundary, and leaf-shaped growth; large internal necrosis of tumour; and infiltration or metastasis of surrounding tissue. This indicates a malignant tumour. The final qualitative diagnosis depends on surgical pathology and immunohistochemistry.

## Data Availability

The data presented in this study are available on request from the corresponding author.

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
