# Peer review of "Gastrointestinal Stromal Tumour with Liver Metastasis Presenting as Gastric Cancer"

_diagnostics, 2023, doi:10.3390/diagnostics13030376_

Round 1

Reviewer 1 Report

Dear Authors

The authors present an interesting case report for GIST of the stomach.

I would like to ask a few questions as follows

1. what are the differences in endoscopic findings between gastric cancer and gastric GIST? For example, is the fact that the rise of the tumor is in the normal mucosa a finding that should raise suspicion of disease other than gastric cancer?

2. GIST has very characteristic pathological findings of HE staining, and histological diagnosis is considered to be very important. From which site and by what method is tissue sampling considered most effective during endoscopy?

Round 1

Response to Reviewer 1 Report

  1. what are the differences in endoscopic findings between gastric cancer and gastric GIST? For example, is the fact that the rise of the tumour is in the normal mucosa a finding that should raise suspicion of disease other than gastric cancer?

Author Response

Thank you for your comments!

Through this case, we can see that there is no big difference between gastric cancer and gastric GIST under endoscopy. Because of the manifestation under endoscopy, we were diagnosed with gastric cancer after endoscopic diagnosis and finally diagnosed the gastric gist by pathological diagnosis. The interesting cause of this case is that the stomach tumour began to consider, and finally diagnosed the gastric gist.

  The tumour is in the normal mucosa a finding that should raise suspicion gastric lymphoma may be considered. But the pathological examination is the gold standard for the diagnosis.

  1. GIST has very characteristic pathological findings of HE staining, and histological diagnosis is considered to be very important. From which site and by what method is tissue sampling considered most effective during endoscopy?

Author Response

Thank you for your comments!

Through the endoscopy, a biopsy of the deep tissue around the lesion. Through endoscopy, deep tissue, multisite biopsy pathology was performed in gastric tissue.

Reviewer 2 Report

This was a case of Gastrointestinal Stromal Tumor with liver metastasis presenting as gastric cancer. Similar cases have been reported, but this may be educational. However, there are several issues which needs consideration.

Comments

Did she have a history of oral administration of NSAIDs, history of Helicobacter pylori infection, HpIgG, or Urea Breath Test results?

 What is the main point the author wants to show in this case? Authors should write their case presentations and discussions along that point.

 In endoscopic findings, the surrounding area appears to have an SMT-like shape. Such observations may also be taken into account in the discussion.

 Figure legends should be limited to the description of the figure, and a general explanation should be included in the text.

Round 1

Response to Reviewer 2 Report

Did she have a history of oral administration of NSAIDs, history of Helicobacter pylori infection, HpIgG, or Urea Breath Test results?

Author Response

Thank you for your comments!

She did not take NSAIDs. The patient was not tested for Hp at our hospital. After a telephone consultation with the patient, HP tested negative in an outside hospital.

 What is the main point the author wants to show in this case? Authors should write their case presentations and discussions along that point.

Author Response

Thank you very much for your advice. I have added the following paragraph to the case presentations:

GIST is the most common tumour of mesenchymal tissue origin originating from the digestive tract, accounting for 0.1%-3% of gastrointestinal malignancies. GIST usually has no specific symptoms, and is often found incidentally during physical examination or surgical treatment of other diseases. Some patients take gastrointestinal bleeding as the primary symptom, mainly manifested as abdominal pain, abdominal mass, nausea, vomiting, fatigue and weight loss. This patient has had intermittent abdominal pain and was unsuitable for 2 years, and the patient has not paid much attention to it. The gastroscopy of the patient suggested the possibility of gastric cancer. In the imaging diagnosis, also presented with gastric malignancy with liver metastasis. The patient almost gave up treatment. Finally, the pathological diagnosis of GIST, and the patient has the hope of treatment.

In endoscopic findings, the surrounding area appears to have an SMT-like shape. Such observations may also be taken into account in the discussion.

Author Response

This is a good suggestion. I have added the following paragraph to the discussion:

In the endoscopic inspection, we found that the surrounding area appears to have an SMT (submucosal tumour, SMT)-like shape. Gastric SMT, refers to the submucosal bulge of the stomach. It is the bulge lesion of the stomach seen under the gastroscope. Because the bulge originates from the submucosa, the colour and structure of the surface are consistent with the surrounding gastric mucosa. Under the endoscopic, we see the stomach occupy, highlight the stomach cavity, show unevenness, and diagnose gastric malignant tumours in endoscopy (Figure 1). The finding of a submucosal bulge in the stomach requires ultrasound gastroscopy to identify the exact source of the submucosa and thus determine its nature. Finally, we performed a pathological examination to diagnose GIST (Figure 4).

 Figure legends should be limited to the description of the figure, and a general explanation should be included in the text.

Author Response

Thanks so much for the reminder. The figure legends have been partially modified.

Round 2

Reviewer 1 Report

The authors answered some of my questions. So, I think this paper could potentially be suitable for publication.

Reviewer 2 Report

I think the text is well‐revised. The author replied point by point.